# Investigating disparity in access to Australian clinical genetic health services for Aboriginal and Torres Strait Islander people

Joanne Luke[1] ✉, Philippa Dalach[1], Lindsay Tuer[2,3], Ravi Savarirayan [2], Angeline Ferdinand[1], Julie McGaughran[4,5], Emma Kowal [6], Libby Massey[3,7], Gail Garvey [8], Hugh Dawkins[9], Misty Jenkins [10], Yin Paradies[11], Glenn Pearson[12], Chloe A. Stutterd[2,13], Gareth Baynam [12,14,15] & Margaret Kelaher[1,16]

Globally, there is a recognised need that all populations should be able to access the benefits of genomics and precision medicine. However, achieving this remains constrained by a paucity of data that quantifies access to clinical genomics, particularly amongst Indigenous populations. Using administrative data from clinical genetic health services across three Australian jurisdictions (states/territories), we investigate disparities in the scheduling and attendance of appointments among Aboriginal and/or Torres Strait Islander people, compared to non-Indigenous people. For 14,870 appointments scheduled between 2014–2018, adjusted Multivariate Poisson Regression models revealed that Aboriginal and/or Torres Strait Islander people were scheduled fewer appointments (IRR 0.73 [0.68–0.80], <0.001) and attended at lower rates (IRR 0.85 [0.78–0.93], <0.001). Within this population, adults, females, remote residents, and those presenting in relation to cancer or prenatal indications experienced the greatest disparity in access. These results provide important baseline data related to disparities in access to clinical genomics in Australia.

[1]Centre for Health Policy, The University of Melbourne, Melbourne, VIC, Australia. [2]Victorian Clinical Genetics Services, Murdoch Children's Research Institute, Parkville, VIC, Australia. [3]Machado Joseph Disease Foundation, Alyangula, NT, Australia. [4]Genetic Health Queensland, Royal Brisbane and Women's Hospital, Brisbane, QLD, Australia. [5]School of Medicine, University of Queensland, St Lucia, QLD, Australia. [6]Alfred Deakin Institute for Citizenship and Globalisation, Deakin University, Geelong, VIC, Australia. [7]James Cook University, Townsville, QLD, Australia. [8]Wellbeing and Preventable Chronic Disease Division, Menzies School of Health Research, Charles Darwin University, Casuarina, NT, Australia. [9]School of Medicine, The University of Notre Dame Australia, Sydney, NSW, Australia. [10]Immunology, Walter Eliza Hall Institute, Parkville, VIC, Australia. [11]School of Humanities and Social Science, Deakin University, Burwood, VIC, Australia. [12]Telethon Kids Institute and Division of Paediatrics, Faculty of Health and Medical Sciences, University of Western Australia, Perth, WA, Australia. [13]Department of Paediatrics, University of Melbourne, Parkville, VIC, Australia. [14]Genetic Services of Western Australia, Western Australian Department of Health, Perth, WA, Australia. [15]Western Australian Register of Developmental Anomalies, Western Australian Department of Health, Perth, WA, Australia. [16]Deceased: Margaret Kelaher. ✉e-mail: jnluke@unimelb.edu.au

Globally there is a robust and growing evidence base that reveals access and outcomes across health systems are different for Indigenous populations. For Aboriginal and/or Torres Strait Islander populations, research reveals disparities in access to the Australian health system and the clinical services it provides, including diagnostic investigations, procedures, care planning, treatments, as well as service adherence to best practice treatment guidelines[1–11]. However, to date, access to clinical genetic health services has not been quantified among Aboriginal and Torres Strait Islander populations. Here we focus on public clinical genetic health services that sit within the broader Australian health system, which has been reported to produce racialised health inequities[12]. This research recognises that similar disparities in access to health services and in health outcomes, are experienced by Indigenous populations globally[13].

Genomic medicine is now recognised as a standard component of health care[14]. Clinical genetic health services are key providers of genomic medicine, a rapidly evolving field that has demonstrated benefits for individuals and families across the life-course[15]. Working with molecular and cytogenic laboratories, clinical genetic health services provide diagnostic assessments and molecular confirmation of disease or disease risk, supported by genetic counselling. In Australia, individuals and families are referred to clinical genetic health services by general practitioners (GPs) (i.e. primary care providers), hospitals and specialists, as well as through self-referral pathways in some jurisdictions. For people with or at risk of genetic conditions, including hereditary cancer syndromes and rare diseases, clinical genetic health services provide substantial health benefits through diagnosis, prevention, medical advice, education, and counselling[16].

Globally, there is a paucity of any data that relates to access to clinical genetic health services and the concomitant benefits they provide. Yet, despite this there is a recognised concern that Indigenous populations do not have equal access to clinical genetic health services and inclusion in genomic research[17,18].

As we consider the need for clinical genetic health services, all available data suggests there is a greater need for services amongst Aboriginal and/or Torres Strait Islander populations. Australian Institute of Health and Welfare data reveals higher rates of incident cancers for Aboriginal and/or Torres Strait Islander populations, suggesting a likely greater need for clinical genetic health services amongst Aboriginal and/or Torres Strait Islander people with cancer and their families[19]. Similarly, the Australian Bureau of Statistics reports higher fertility rates for Aboriginal and or/Torres Strait Islander women, who, across their lifetime, have an average of 2.32 babies per woman compared to 1.66 babies per non-Aboriginal and or/Torres Strait Islander woman, which also suggests a greater need for clinical genetic health services in the prenatal period[20]. A younger age distribution for the Aboriginal and/or Torres Strait Islander population also suggests we should expect a greater frequency of rare diseases amongst Aboriginal populations, as rare diseases are mostly diagnosed early in life[21]. Further, as we consider that the term 'Aboriginal and/or Torres Strait Islander' is a social construct which includes people of great biological heterogeneity, there is no prima facie reason to expect lower susceptibility to either rare genetic conditions in pregnancy or hereditary cancer for Aboriginal and Torres Strait Islander people. All these factors combined, allude to a greater need for clinical genetic health services rather than equal need for Aboriginal and/or Torres Strait Islander people.

Qualitative studies have also shown that Aboriginal and Torres Strait Islander people wish to be included in the benefits offered by genomic medicine by having access to clinical genetic health services. Wild et al. (2013) found that there was demand among Aboriginal women for genetic investigations and counselling in the antenatal period and Bernardes et al. (2014) similarly showed that many Aboriginal and Torres Strait Islander people with cancer were interested in discussing with a genetic specialist the subsequent risk of cancer among family members[22,23]. Australian health policy has further recognised the importance of identifying and addressing barriers to genetic health services for Aboriginal and Torres Strait Islander people. The National Health Genomics Policy Framework and Implementation Plan 2018–2021, states as part of its mission that the integration of genomics into the health system must proceed in 'an efficient, effective, ethical and equitable way'[24]. The Framework specifically targets Aboriginal and Torres Strait Islander peoples in three priority areas for action: evaluating the accessibility, appropriateness and cultural responsiveness of clinical genetic health services, increasing genomic literacy of health providers working with Indigenous populations and ensuring culturally safe data collection in order to reflect diversity[24].

Responding to these factors, the aim of this research is to investigate access to clinical genetic health services by examining rates of appointment scheduling and attendance among Aboriginal and/or Torres Strait Islander people in three Australian jurisdictions: the Northern Territory, Western Australia and Queensland. In these jurisdictions Aboriginal people retrospectively comprise 30.3%, 4.6% and 3.9% of the resident population, and these proportions at a minimum should be reflected in referral rates. This research makes an essential contribution to the field of health disparities research as it is the first study globally that quantitatively reports access to clinical genetic health services amongst an Indigenous population.

## Results

In total, 17,217 people had an appointment scheduled across the three clinical genetic health services for the years examined (2014–2018). Of these, 14,870 people had Aboriginal and/or Torres Strait Islander status recorded and are included in these analyses.

Of the total 4,285 appointments scheduled annually, 3.4% were scheduled at the Northern Territory Genetics Service, 56.2% at Genetic Services of Western Australia, and 40.4% at Genetic Health Queensland. Aboriginal and/or Torres Strait Islander people comprised 20.4%, 3.3% and 4.2% of all scheduled appointments in Northern Territory, Western Australia, and Queensland respectively, slightly lower than the expected 30.3%, 3.9% and 4.6% we would expect based on population distribution.

In total, 4.4% of all scheduled appointments were for people who identified as Aboriginal and/or Torres Strait Islander people across the three states. Based on population parity we expected 5.2% of appointments to be for Aboriginal and/or Torres Strait Islander people in combined tristate data. The subsequent analyses describe disparity in appointment scheduling and attendance for these people.

### Characteristics of people for whom an appointment was scheduled

Aboriginal and/or Torres Strait Islander people who were scheduled an appointment were younger, less likely to reside in major cities and less likely to be referred by a specialist in comparison to non-Aboriginal and/or Torres Strait Islander people (Table 1). Gender ratios differed for Aboriginal and/or Torres Strait Islander people relative to non-Aboriginal and/or Torres Strait Islander people. Aboriginal and/or Torres Strait Islander people were more likely to have a telehealth appointment overall. However, excluding people from major cities, Aboriginal and/or Torres Strait Islander people from regional and remote areas were less likely to have a telehealth appointment (9% vs 14%).

In terms of reasons for the appointment, a greater proportion of Aboriginal and/or Torres Strait Islander people were scheduled an appointment for a rare disease than for cancer and prenatal reasons in comparison to non-Aboriginal and/or Torres Strait Islander people. For Aboriginal and/or Torres Strait Islander people, the rate of scheduling appointments for prenatal reasons was 72% lower, with only two

**Table 1 | Characteristics of people scheduled an appointment, by Aboriginal and/or Torres Strait Islander status**

| | Aboriginal and/or Torres Strait Islander people (annual n = 182) | | | non-Aboriginal and/or Torres Strait Islander people (annual n = 4103) | | | |
|---|---|---|---|---|---|---|---|
| | n | (%) | 95%CI | n | (%) | 95%CI | p-value* |
| **Age** | | | | | | | |
| 0–9 | 83 | 45.7 | (38.5–53.0) | 859 | 20.9 | (19.7–22.2) | <0.001 |
| 10–19 | 28 | 15.6 | (10.3–20.8) | 355 | 8.6 | (7.8–9.5) | |
| 20–29 | 21 | 11.6 | (6.9–16.2) | 531 | 12.9 | (11.9–14.0) | |
| 30–39 | 20 | 10.8 | (6.3–15.3) | 788 | 19.2 | (18.0–20.4) | |
| 40–49 | 11 | 6.2 | (2.7–9.7) | 589 | 14.4 | (13.3–15.4) | |
| 50+ | 18 | 10.0 | (5.6–14.3) | 981 | 23.9 | (22.6–25.2) | |
| **Gender** | | | | | | | |
| Female | 103 | 56.5 | (49.2–63.6) | 2733 | 66.7 | (65.2–68.1) | <0.001 |
| Male | 79 | 43.5 | (36.1–50.5) | 1363 | 33.3 | (31.8–34.7) | |
| **Remoteness** | | | | | | | |
| Major city | 70 | 40.5 | (33.1–47.8) | 3062 | 77.1 | (75.8–78.4) | <0.001 |
| Regional | 66 | 38.2 | (30.9–45.4) | 776 | 19.5 | (18.3–20.8) | |
| Remote | 37 | 21.4 | (15.3–27.5) | 135 | 3.4 | (2.8–4.0) | |
| **Referral by** | | | | | | | |
| GP | 31 | 19.8 | (13.6–26.1) | 1124 | 31.4 | (29.9–32.9) | <0.001 |
| Specialist | 107 | 69.0 | (61.8–76.4) | 1950 | 54.4 | (52.8–56.0) | |
| Other | 17 | 11.2 | (6.3–16.2) | 510 | 14.2 | (13.1–15.4) | |
| **Location of service** | | | | | | | |
| Clinic | 129 | 94.0 | (90.0–98.0) | 3991 | 97.3 | (96.8–97.8) | <0.001 |
| Telehealth | 8 | 5.8 | (1.9–9.8) | 112 | 2.7 | (2.2–3.2) | |
| **Reason** | | | | | | | |
| Rare disease | 109 | 75.7 | (68.7–82.7) | 1796 | 56.0 | (54.3–57.7) | <0.001 |
| Cancer | 35 | 24.3 | (17.3–31.3) | 1410 | 44.0 | (42.3–45.7) | <0.001 |
| Prenatal | 1 | 0.7 | (0–2) | 125 | 3.1 | (2.5–3.6) | <0.001 |

*Pearson $\chi^2$ test.

appointments scheduled annually for Aboriginal and/or Torres Strait Islander people across the three states.

Characteristics of Aboriginal and/or Torres Strait Islander people who were scheduled an appointment at a clinical genetic health service differed by state and are described in Table 2. In the Northern Territory, most people were under 20 years of age (77.7%) and most were scheduled an appointment for a rare disease. Conversely, in Queensland, more Aboriginal and/or Torres Strait Islander people were scheduled an appointment at an older age, although half (49.1%) were still aged under 20 years. In Queensland one in three appointments scheduled was for cancer, compared to one in ten appointments in the Northern Territory and one in five in Western Australia. In Queensland, unlike the Northern Territory and Western Australia, there was an observed gender differential among Aboriginal and/or Torres Strait Islander people, where females were more likely to be scheduled an appointment than males.

Darwin and Alice Springs (the two largest cities in the Northern Territory) are classified under the ASGS remoteness structure as outer regional and remote, respectively. As a result, no individuals in the Northern Territory lived in major cities. People in the Northern Territory were more likely to live in regional (54.5%) and remote (45.5%) areas, while in Western Australia and Queensland, half of appointments scheduled were for people living in major cities (i.e. Perth and Brisbane). In Western Australia 26.3% of scheduled appointments were for people residing in remote areas, while for Queensland this figure was 4.9%.

A vast majority of referrals in the Northern Territory came from a specialist (94.6%), whereas in fewer referrals came from a specialist in Western Australia (62.5%) and Queensland (63.45%), with the rest from a general practitioner, other clinician or by self-referral.

### Investigating disparity in appointment scheduling

Based on univariate modelling, incidence of appointment scheduling was 46 per 100,000 Aboriginal and/or Torres Strait Islander people annually compared to 57 per 100,000 for non-Aboriginal and/or Torres Strait Islander people. This reflects an under-scheduling rate of 19.2% (Table 3). If there were parity in appointment scheduling, we would have expected to see 42 additional appointments scheduled annually for Aboriginal and/or Torres Strait Islander people across the three states. Under-scheduling for Aboriginal and/or Torres Strait Islander people was greatest amongst adults, people residing in remote areas, and females. The greatest disparity was observed in the Northern Territory, where incidence scheduling was 40 per 100,000 for Aboriginal and/or Torres Strait Islander people compared to 67 per 100,000 for non-Aboriginal and/or Torres Strait Islander people.

There did not appear to be disparity in appointment scheduling for rare diseases (excluding prenatal appointments as a subcategory of rare disease). However, for cancer and prenatal reasons, scheduled appointments were 55.8% and 71.4% lower for Aboriginal and/or Torres Strait Islander people.

### Characteristics of people attending an appointment

Annually, 75.4% of Aboriginal and/or Torres Strait Islander people attended a scheduled appointment compared to 86.5% of non-Aboriginal and/or Torres Strait Islander people (Table 4).

For Aboriginal and/or Torres Strait Islander people, attendance was lowest for people in the 20–29 year age group, and males. Unlike non-Aboriginal and/or Torres Strait Islander people, where attendance decreased with remoteness, attendance for Aboriginal and/or Torres Strait Islander people residing in a major-city (71.1%) and remotely (72.9%) were similar, reflecting that attendance was low irrespective of geographic reach. The reason for the appointment did not impact attendance. All prenatal appointments were attended. However, caution must be used in interpretation of this finding as there were only two prenatal appointments annually for Aboriginal and/or Torres Strait Islander people.

Attendance overall was lowest in Western Australia, with 65.0% of Aboriginal and/or Torres Strait Islander people attending (compared to 76.7% in the Northern Territory and 86.1% in Queensland) and 79.6% of non-Aboriginal and/or Torres Strait Islander people attending (compared to 94.3% in the Northern Territory and 95.6 % in Queensland). Rate ratios for each jurisdiction reveal that despite differences in attendance by state, rate ratio odds were consistently 18% lower for Aboriginal and/or Torres Strait Islander people compared to non-Aboriginal and/or Torres Strait Islander people.

### Investigating disparity in appointment scheduling and attendance-multivariate analysis

In multivariate analysis adjusting for age, gender and state, IRR revealed that Aboriginal and/or Torres Strait Islander people were 27% less likely to be scheduled an appointment at a clinical genetic health service and attended appointments at a 15% lower rate than non-Aboriginal and/or Torres Strait Islander people ($p < 0.001$, Table 5).

## Discussion

This paper is the first to quantify access to and use of clinical genetic health services for an Indigenous population across multiple jurisdictions within a country. Our analyses provide clear evidence for marked disparity in access for Aboriginal and/or Torres Strait people, both in terms of appointment scheduling and attendance at clinical genetic health services across three jurisdictions. These findings are of

**Table 2 | Characteristics of Aboriginal and/or Torres Strait Islander people scheduled an appointment, by State**

| | Northern Territory (2014–2018) *n* = 148 (annual *n* = 30) | | Western Australia (2015–18) *n* = 320 (annual *n* = 80) | | Queensland (2015–17) *n* = 216 (annual *n* = 72) | | Total (annual = 182) | | | |
|---|---|---|---|---|---|---|---|---|---|---|
| | % | (95%CI) | % | (95%CI) | % | (95%CI) | *n* | (%) | 95%CI | *p*-value* |
| **Age** | | | | | | | | | | |
| 0–9 | 62.2 | (54.3–70.0) | 50.6 | (45.1–56.1) | 33.8 | (27.5–40.1) | 83 | 45.7 | (38.5–53.0) | <0.001 |
| 10–19 | 15.5 | (9.7–21.4) | 15.9 | (11.9–19.9) | 15.3 | (10.5–20.1) | 28 | 15.6 | (10.3–20.8) | |
| 20–29 | 10.1 | (5.3–15.0) | 9.7 | (6.4–12.9) | 14.4 | (9.7–19.0) | 21 | 11.6 | (6.9–16.2) | |
| 30–39 | 5.4 | (1.8–9.0) | 12.5 | (8.9–16.1) | 11.1 | (6.9–15.3) | 20 | 10.8 | (6.3–15.3) | |
| 40–49 | 6.1 | (2.2-9.9) | 4.7 | (2.4–7.0) | 7.9 | (4.3–11.5) | 11 | 6.2 | (2.7–9.7) | |
| 50+ | 0.7 | (0–2.0) | 6.6 | (3.8–9.3) | 17.6 | (12.5–22.7) | 18 | 10.0 | (5.6–14.3) | |
| **Gender** | | | | | | | | | | |
| Female | 54.4 | (46.4–62.5) | 53.8 | (48.1–59.0) | 60.6 | (54.1–67.2) | 103 | 56.4 | (49.2–63.6) | 0.259 |
| Male | 45.6 | (37.5–53.6) | 46.3 | (41.0–51.9) | 39.4 | (32.8–45.9) | 79 | 43.3 | (36.1–50.5) | |
| **Remoteness** | | | | | | | | | | |
| Major city | | | 50.0 | (44.4–55.6) | 47.5 | (40.7–54.4) | 70 | 40.7 | (33.3–48.0) | <0.001 |
| Regional | 54.5 | (46.4–62.6) | 23.7 | (18.9–28.5) | 47.5 | (40.7–54.4) | 66 | 38.2 | (31.0–45.5) | |
| Remote | 45.5 | (37.4–53.6) | 26.3 | (21.4–31.3) | 4.9 | (1.9–7.9) | 37 | 21.1 | (15.0–27.2) | |
| **Referral by** | | | | | | | | | | |
| GP | 4.1 | (0.9–7.2) | 10.1 | (6.3–13.8) | 36.6 | (29.8–43.5) | 31 | 16.9 | (11.5–22.4) | <0.001 |
| Specialist | 94.6 | (91.0–98.2) | 62.5 | (56.5–68.5) | 63.4 | (56.5–70.2) | 107 | 58.8 | (51.7–66.0) | |
| Other | 1.4 | (0–3.2) | 27.4 | (21.9–33.0) | - | | 17 | 9.6 | (5.3–13.8) | |
| **Reason** | | | | | | | | | | |
| Rare disease | 88.5 | (83.2–93.8) | 80.9 | (75.2–86.5) | 67.5 | (61.1–73.9) | 109 | 59.9 | (52.7–67.0) | <0.001 |
| Prenatal | 0.7 | (0–2.0) | 0.3 | (0–0.9) | 1.9 | (0.1–3.6) | 2 | 1.0 | (0–2.4) | 0.156 |
| Cancer | 11.5 | (6.2–16.8) | 19.1 | (13.5–24.8) | 32.5 | (26.1–38.9) | 35 | 19.0 | (13.3–24.7) | <0.001 |

*Pearson $\chi^2$ test.

importance to clinical genetic health services within Australia, as well as to health services globally as they consider and advance access to genomic and precision medicine.

To contextualise these findings, we draw on the literature relating to disparities in access to clinical genetic health service for marginalised populations in Australia, New Zealand, Canada and the US. This literature has largely taken an ecological perspective and a qualitative approach to describe many factors contributing to health disparities at the individual, interpersonal, health system levels (Fig. 1).

At the health system and society level the literature highlights policy and practice mechanisms contributing to disparities including the Eurocentric biomedical health model with its emphasis on individualism and autonomy, the pro-white bias (or whiteness) of delivery and standardisation of care, as well as geographical proximity to services[14,25]. At the interpersonal level–that is, the patient-practitioner interface–the literature identifies the possible contribution of referring practitioners and genetic health practitioners to disparity[22,26–28]. At the individual level, this literature speaks of individual attributes such as awareness, knowledge, attitudes, values, preferences, and priorities that compromise access, and which are shaped by the socio-cultural context[25,29–32]. These disparities are also exacerbated by the high non-medical costs associated with attending services, particularly for patients in rural and remote settings travelling long distances to access services. Addressing disparities in access to clinical genetic health services is likely to involve intervention across each of these levels.

As we attempt to better understand disparities in access to Australian clinical genetic health services, we draw on a health equity lens. Through a health equity lens, health inequalities are seen as the unjust disparities in access to health resources that arise from the socio-cultural-historical-political context that different population groups live. 'Through a health equity lens we bring attention to the historic

and ongoing racism experienced by Aboriginal and Torres Strait Islander peoples in Australian health settings reported more broadly[33,34]. Through this lens, access to health services for Aboriginal and/or Torres Strait Islander people is understood within an ecological frame that is shaped by spatial and temporal aspects that cannot be separated from past and ongoing processes of colonialism[34–36]. We acknowledge how research and clinical practices have made inappropriate and unethical use of Aboriginal and Torres Strait Islander peoples' genetic information in the past, as well as recognise that Australian medical research practices and health service delivery models were developed by and for the dominant culture, with minimal consideration of cultural safety and responsiveness to the needs of Aboriginal and Torres Strait Islander communities[37,38]. We also consider the role that medical professionals, researchers, and policy makers have played in Australia and globally in the construction of dehumanising and racialised knowledge that claimed to show that Indigenous minds, bodies and cultures were inferior or less fit[36–39]. In particular, it was Australian state governments that drew on racial science, including the pseudoscience of eugenics, in their attempt to eradicate and 'breed out' Aboriginal and Torres Strait Islander people, including through official assimilation and segregation policies[40–42]. We bring attention to these socio-cultural-historical-political contexts of Aboriginal and/or Torres Strait Islander people's lives, as health disparities for Indigenous populations are unlikely devoid of context[43]. However, identifying the nature of relationships between these contexts and health disparities is the work of future research led and in partnership with Aboriginal and/or Torres Strait Islander people.

We found significant under-scheduling of appointments for Aboriginal and/or Torres Strait Islander people, with disparities most marked for those from remote areas, adults, and those referred for cancer or during the prenatal period. Under-scheduling of

**Table 3 | Disparity in annual appointment scheduling for Aboriginal and/or Torres Strait Islander people (univariate analysis)**

| | Aboriginal and/or Torres Strait Islander | | Non Aboriginal and/or Torres Strait Islander | | Difference Rate ratio (95% CI) | | Aboriginal and/or Torres Strait Islander Expected referral (n) | Aboriginal and/or Torres Strait Islander under–referred (%) |
|---|---|---|---|---|---|---|---|---|
| | *n* | Incident referral per 100,000 (95%CI) | *n* | Incident referral per 100,000 (95%CI) | | | | |
| **Age group (years)** | | | | | | | | |
| 0–9 | 83 | 90 (71–110) | 859 | 92 (86–98) | 0.98 | (0.79–1.23) | 84 | 1.2 |
| 10–19 | 28 | 34 (22–47) | 355 | 41 (37–45) | 0.83 | (0.57–1.22) | 34 | 17.6 |
| 20–29 | 21 | 31 (18–44) | 531 | 51 (47–55) | 0.60 | (0.39–0.92) | 35 | 40.0 |
| 30–39 | 20 | 40 (22–57) | 788 | 77 (71–82) | 0.52 | (0.34–0.82) | 38 | 47.4 |
| 40–49 | 11 | 25 (10–40) | 589 | 59 (54–64) | 0.42 | (0.23–0.76) | 26 | 57.7 |
| 50+ | 18 | 31 (17–45) | 981 | 41 (39–44) | 0.75 | (0.47–1.19) | 24 | 25.0 |
| **Gender** | | | | | | | | |
| Female | 103 | 52 (42–61) | 2733 | 75 (72–78) | 0.69 | (0.57–0.84) | 149 | 30.9 |
| Male | 79 | 40 (11–36) | 1363 | 38 (36–40) | 1.06 | (0.85–1.33) | 74 | −6.8 |
| **Remoteness** | | | | | | | | |
| Major city | 70 | 61 (46–75) | 3062 | 62 (60–64) | 1.02 | (0.81–1.29) | 72 | 2.8 |
| Regional | 66 | 44 (34–55) | 776 | 38 (35–41) | 0.85 | (0.66–1.10) | 56 | −17.9 |
| Remote | 37 | 38 (19–37) | 135 | 54 (45–63) | 1.92 | (1.33–2.76) | 71 | 47.9 |
| **State** | | | | | | | | |
| NT | 30 | 40 (26–55) | 115 | 67 (55–79) | 0.60 | (0.40–0.89) | 50 | 40.0 |
| WA | 80 | 80 (62–97) | 2330 | 95 (91–99) | 0.84 | (0.67–1.05) | 95 | 15.8 |
| QLD | 72 | 33 (25–40) | 1658 | 36 (34–38) | 0.91 | (0.72–1.15) | 79 | 8.9 |
| **Reason** | | | | | | | | |
| Rare disease | 109 | 27 (22–33) | 1796 | 25 (24–26) | 1.11 | (0.92–1.35) | 98 | −11.2 |
| Cancer | 35 | 9 (6–12) | 1410 | 19 (18–20) | 0.45 | (0.32–0.64) | 77 | 54.5 |
| Prenatal | 2 | 1 (0–3) | 125 | 4 (4–5) | 0.28 | (0.07–1.14) | 7 | 71.4 |
| TOTAL | 182 | 46 (39–52) | 4103 | 57 (55–58) | 1.23 | (1.06–1.43) | 224 | 19.2 |

appointments in remote areas is likely reflective of a referral bias, where opportunity for referral is restricted by the limited availability of both general practitioners and specialist services in remote areas[44]. It may also reflect a relative lack of awareness of genetic and rare diseases in primary care when compared to specialist care. This speaks to a need for increased awareness in primary care and enhanced referral pathways in the absence of general practice and specialist services (including self-referral to clinical genetic health services). Additionally, the relative lack of telehealth appointments may reflect a requirement to increase and tailor provision of telehealth services to meet the needs of Aboriginal and Torres Strait Islander people. This has become acutely apparent in the provision of other health services during the COVID-19 pandemic[45]. There may also be specific considerations for improving telehealth for genetic health care that cater to the frequently familial nature of diseases and consultations, and support confidentiality and culturally appropriate mechanisms of gaining informed consent.

Across jurisdictions there was under-scheduling of appointments for adults, which could be explained by low rates of prenatal and cancer appointments. Given that both Aboriginal and Torres Strait Islander and non-Aboriginal and Torres Strait Islander populations are heterogenous and that there is no prima facie reason to expect lower susceptibility to either rare genetic conditions in pregnancy or hereditary cancer in Aboriginal and Torres Strait Islander people, it is unlikely that this discrepancy in access reflects a biological driver. Indeed Allford et al. (2014), drawing on a study by Hall et al. (2009), highlight that mutations in the Mendelian genes that confer hereditary susceptibility to cancers occur in equal frequency across different ethnic groups[46]. Further, as the Australian Institute of Health and Welfare reported in 2018 using national morbidity data, Aboriginal and Torres Strait Islander populations were 10% more likely to be

diagnosed with cancer in 2009–2013, which we would have expected to precipitate a higher, not lower, rate of appointments scheduled for Aboriginal and Torres Strait Islander people[19]. Also, consistent with our findings, a review of studies from the United Kingdom, North America and Australasia revealed lower rates of access to services for cancer among minority social groups[46]. We recognise that cancer discourse relating to Aboriginal and/or Torres Strait Islander people is consistently framed as a consequence of individual deficits in behaviours and lifestyle choices (such as smoking), and this stigmatisation may also contribute to under-referral[47].

In terms of prenatal appointment scheduling, only one appointment was scheduled for Aboriginal and/or Torres Strait Islander people across the three states per year; a rate 72% lower than non-Aboriginal and/or Torres Strait Islander people. However, based on higher fertility rates, we would expected to see more appointments scheduled for Aboriginal and/or Torres Strait Islander women. Arabena (2006) highlights the importance of sexual and reproductive rights as a means to improve the lives of Aboriginal and/or Torres Strait Islander people[48]. These sexual and reproductive rights include access to services and information, including prenatal screening and testing as well as reproductive genetic counselling for people at risk preconception. It must also be considered that for Aboriginal and/or Torres Strait Islander women their experiences of medical care continues to be impacted by past policies of forced child removal and sterilisation[49].

In this paper, we have likely underestimated the actual disparity rates of Aboriginal and/or Torres Strait Islander people accessing clinical genetic health services. Our interpretation of data is based on assumed equal need for Clinical Genetic Health Services by both groups. However, based on the higher rates of incident cancer for Aboriginal and Torres Strait Islander people and higher rates of fertility

**Table 4 | Disparity in annual attendance for Aboriginal and/or Torres Strait Islander people (univariate analysis)**

| | Aboriginal and/or Torres Strait Islander | | | Non Aboriginal and/or Torres Strait Islander | | | Difference | |
|---|---|---|---|---|---|---|---|---|
| | n | % | 95%CI | n | % | 95%CI | % difference | Rate ratio (95% CI) |
| **Age group (years)** | | | | | | | | |
| 0–9 | 63 | 76.3 | (67.1–85.4) | 720 | 83.8 | (81.4–86.3) | 7.6 | 0.91 (0.80–1.03) |
| 10–19 | 20 | 70.1 | (53.2–86.9) | 289 | 81.6 | (77.5–85.6) | 11.5 | 0.86 (0.69–1.11) |
| 20–29 | 14 | 68.1 | (48.2–88.0) | 465 | 87.5 | (84.7–90.4) | 19.5 | 0.76 (0.56–1.03) |
| 30–39 | 16 | 79.4 | (61.5–97.3) | 690 | 87.5 | (85.2–89.9) | 8.1 | 0.91 (0.73–1.14) |
| 40–49 | 9 | 76.7 | (51.9–100) | 521 | 88.5 | (85.9–91.0) | 11.8 | 0.85 (0.61–1.18) |
| 50+ | 15 | 80.3 | (65.7–100) | 863 | 87.9 | (85.9–90.0) | 4.9 | 0.95 (0.77–1.17) |
| **Gender** | | | | | | | | |
| Female | 80 | 78.2 | (70.2–86.2) | 2376 | 86.9 | (85.7–88.2) | 8.7 | 0.90 (0.81–1.00) |
| Male | 57 | 71.6 | (61.7–81.5) | 1167 | 85.6 | (83.8–87.5) | 14.0 | 0.80 (0.69–0.91) |
| **Remoteness** | | | | | | | | |
| Major city | 51 | 71.1 | (65.5–76.7) | 2649 | 86.5 | (85.3–87.7) | 13.7 | 1.19 (1.03–1.37) |
| Regional | 52 | 79.3 | (69.5–89.0) | 666 | 85.9 | (83.5–88.4) | 6.7 | 1.09 (0.96–1.24) |
| Remote | 27 | 72.9 | (58.4–87.3) | 108 | 80.0 | (73.2–86.7) | 7.1 | 1.10 (0.89–1.36) |
| **State** | | | | | | | | |
| NT | 30 | 76.7 | (61.5–91.8) | 108 | 94.3 | (90.0–98.5) | 17.6 | 0.82 (0.67–1.00) |
| WA | 80 | 65.0 | (54.5–75.5) | 1855 | 79.6 | (78.0–81.2) | 14.6 | 0.82 (0.67–1.00) |
| QLD | 72 | 86.1 | (78.1–94.1) | 1585 | 95.6 | (94.6–96.6) | 9.5 | 0.82 (0.69–0.96) |
| **Reason** | | | | | | | | |
| Rare disease | 85 | 78.2 | (70.4–85.9) | 1562 | 87.0 | (85.4–88.5) | 8.8 | 0.90 (0.81–0.99) |
| Cancer | 26 | 74.9 | (60.4–89.4) | 1261 | 89.4 | (87.8–91.0) | 14.5 | 0.83 (0.68–1.01) |
| Prenatal | 2 | 100.0 | | 114 | 98.6 | (96.5–100) | –1.4 | 1.10 (1.04–1.16) |
| Total attended | 137 | 75.4 | (69.1–81.6) | 4103 | 86.5 | (85.4–87.5) | 11.1 | 0.87 (0.80–0.95) |

amongst Aboriginal and Torres Strait Islander women, we should have seen incident appointment scheduling rates greater for Aboriginal and/or Torres Strait Islander people rather than equal. Further work is required to quantify with greater precision the actual need and want for clinical genetic health services by Aboriginal and/or Torres Strait Islander people. However, it is likely that policy responses to increase access to genetic health services will need to target a higher rate of uptake for Aboriginal and/or Torres Strait Islander populations rather than equal uptake for both populations.

It is encouraging that our data reveals that Aboriginal and/or Torres Strait Islander children in the 0–9 and 10–19 year age groups are being scheduled appointments with equal frequency to non-Aboriginal and/or Torres Strait Islander children. One underlying factor for the greater likelihood of rare diseases being referred during childhood, when there is more frequent contact with the health system, thereby providing more opportunity to receive a referral. Also based on the Northern Territory experience, there has been an ongoing policy and practice focus on paediatric care initiatives, including outreach by paediatric teams. Exploring such factors that have supported the apparent relative equal referral for paediatric clinical genetic care provision will provide insights for adult service provision.

Attendance was lowest for Aboriginal and Torres Strait Islander people in the 20–29 year age group, people identified as male and for appointments made in relation to cancer. These groups would likely benefit from further support to attend appointments. As non-attendance includes an appointment being cancelled, moved, or not attended we were unable to ascertain if attendance reflects individual or service-level factors. For age, the lower attendance for those aged 20–29 may reflect competing social responsibilities (caring for children, work, family responsibilities) impacting this age group, which may require rescheduling and cancelling of appointments. Younger adults may not be as familiar with or engaged in navigating health services, which could also impact attendance. With regard to gender, there was lower attendance for people identified as male in both populations, which is consistent with the findings of other research[50]. In interpreting this finding, we draw on the work of Canuto et al. (2018) who found a high level of motivation amongst adult Aboriginal and/or Torres Strait Islander males to attend preventative health care, but that logistical factors, lack of promotion of services, inadequate communication, and lack of culturally appropriate and gender-specific services were barriers to health service utilisation[51]. Our finding of lower rates of attendance at clinical genetic health services for cancer-related appointments may reflect variations in care coordination between disease domains and age groups. We did not find there to be differences in attendance among Aboriginal and/or Torres Strait Islander people based on remoteness, suggesting that the overall attendance disparity observed between remote and non-remote patients relates to differences in rates of referral.

We also found variation in scheduling and attendance at the jurisdictional level. For example, In Queensland, there were significantly more referrals that came from GPs as a proportion of all referrals, which may contribute to the higher referral rates for adults in comparison with other jurisdictions. In Queensland, we also found an observed gender differential where all females were more likely to be scheduled an appointment than males. Given the greater rate of referrals for prenatal reasons and cancer in Queensland, this gender differential may be indicative of higher referral during pregnancy as well as for cancers, many of which are for females ascertained by a family history of breast cancer. Yet for all states, despite their geographical differences and different service models, attendance was consistently 18% lower among Aboriginal and/or Torres Strait Islander people. Understanding the factors outside of those measurable here that underlie the differences between jurisdictions will provide further insights for improvements.

**Table 5 | Disparity in appointment scheduling, multivariate analysis using Poisson regression**

| | Incident Risk ratio | 95%CI | p-value* |
|---|---|---|---|
| **Appointment scheduling**** | | | |
| **Aboriginal and/or Torres Strait Islander person** | | | |
| No (ref) | 1.00 | | |
| Yes | 0.73 | (0.68–0.80) | <0.001 |
| Male | 0.51 | (0.50–0.53) | <0.001 |
| **Age group (years)** | | | |
| 0–9 (ref) | 1.00 | | |
| 10–19 | 0.43 | (0.41–0.46) | <0.001 |
| 20–29 | 0.50 | (0.48–0.53) | <0.001 |
| 30–39 | 0.74 | (0.70–0.77) | <0.001 |
| 40–49 | 0.57 | (0.54–0.60) | <0.001 |
| 50+ | 0.41 | (0.39–0.43) | <0.001 |
| **State** | | | |
| Queensland (ref) | 1.00 | | |
| Northern Territory | 2.85 | (2.63–3.08) | <0.001 |
| Western Australia | 3.47 | (3.35–3.59) | <0.001 |
| **Attendance** | | | |
| **Aboriginal and/or Torres Strait Islander person** | | | |
| No (ref) | 1.00 | | |
| Yes | 0.85 | (0.78–0.93) | <0.001 |
| Male | 1.00 | (0.96–1.04) | <0.952 |
| **Age group (years)** | | | |
| 0–9 (ref) | 1.00 | | |
| 10–19 | 0.95 | (0.89–1.02) | <0.138 |
| 20–29 | 1.00 | (0.94–1.07) | <0.935 |
| 30–39 | 1.01 | (0.96–1.07) | <0.723 |
| 40–49 | 1.00 | (0.94–1.07) | <0.918 |
| 50+ | 1.00 | (0.95–1.05) | <0.894 |
| **State** | | | |
| Queensland | 1.00 | | |
| Northern Territory | 0.98 | (0.90–1.06) | <0.592 |
| Western Australia | 0.83 | (0.80–0.86) | <0.001 |

*$\chi^2$ test.
**Multilevel Poisson regression model weighted for population age distributions (Appointment scheduling only).

The strengths of this cross-sectional study include the relatively large number of people in the study and its inclusion of several jurisdictions within a single country. The numbers provide statistical power, the multi-jurisdictional nature allows for comparisons between service models and the national nature provides some bounds from within which to consider the variability.

Limitations of our analyses are that the data we draw from are administrative appointment databases that have common features, but also differences in data structure, which limits the number and type of factors that can be assessed. Disparities in access can occur at multiple points in a patient journey, including before, during and after an episode of clinical service. The design of this study provides the greatest insights at intermediate and later stages of the patient journey. Further research would be useful to provide insight into drivers of lower referral rates among Aboriginal and Torres Strait Islander people.

We also recognise that Aboriginal and Torres Strait Islander status was missing for one in five people from one jurisdiction (WA). This provides a significant opportunity for improved data collection to enable targeted provision of culturally responsive care and has been

acknowledged as a key objective at both the service- and system-level in this jurisdiction as a result of our findings.

The cross-sectional nature of this analysis means that although we can demonstrate disparity in appointment scheduling and attendance, we cannot definitively determine causation, nor what is driving disparities. Nor can we make definitive conclusions on the quality of the individual episodes of care.

Our findings of disparities in access highlight the need to consider alternative pathways and models of care to improve access for Aboriginal and/or Torres Strait Islander people. Elsum et al. (2020) describe a model of genetic care which has achieved improved access to genomic medicine for Aboriginal people in remote parts of the Northern Territory. This model highlights the benefits of a implementing a community-based, person- and family-centred approach[52]. The family-centred approach to clinical genetic health service provision has also been identified as an important enabler among Māori, the Indigenous people of Aotearoa (New Zealand)[14].

The analyses presented herein quantitively identify key areas for consideration for improved access to clinical genetic health services by Aboriginal and/or Torres Strait Islander people across three Australian states. These findings are likely to provide insights for other Australian jurisdictions and other countries. The global literature also highlights various points at the individual, interpersonal and system levels at which disparities in clinical genetic health service access may be addressed. A study by Reilly et al. (2018) that examined access to clinical cancer services by Aboriginal people found that access improved using a multilevel approach to co-ordinated care. This approach had an emphasis on navigating the health system, providing appropriate information and communication, assisting with the management of multiple and competing social stressors in the home, and was underpinned by cultural safety[53]. These learnings are likely transferable to the clinical genetic health setting. Further, Elsum et al. (2020) in talking to clinical genetic health services highlight the important need for co-design approaches to ensure that health services are both accessible and culturally safe[52].

The analysis revealed there were groups who were shown to be under-represented in appointment scheduling and attendance who would benefit from strategic investment, including adults requiring services during the prenatal period, adults and their family members diagnosed or at risk of cancer, people in remote areas, and people attending primary care irrespective of geographical location.

In considering the literature and our findings we recognise that improving access to clinical genomic health services requires a shift from service design for to with Aboriginal and Torres Strait Islander peoples. There is a critical need for clinical genetic health services and researchers to work with individuals, families and communities to understand where disparities lie, why they arise, and how to work in partnership to design more accessible services[14]. Recognition of this in recent years by clinical genetic health services involved in this study under the 'Achieving Equity in Genetic Health for Indigenous Australians' initiative has seen much work with Aboriginal and Torres Strait Islander people, families, communities and organisations to identify unmet needs and implement more culturally responsive care. Collectively, across the jurisdictions this has included increasing awareness of the need for better linkages to Aboriginal and Torres Strait Islander primary care organisations, increased use of Aboriginal and Torres Strait Islander liaison officers, dedicated roles for Aboriginal and Torres Strait Islander genetic counselling, employing Aboriginal and Torres Strait Islander administrative staff, partnering genetic counsellors to Aboriginal and Torres Strait Islander health and community workers, building genetic and rare diseases education into Aboriginal and Torres Strait Islander Health Care Worker training, making clinics more culturally appropriate with the use of art, flags and language names for clinic rooms and translation of clinical resources into Aboriginal and Torres Strait Islander languages and using Aboriginal

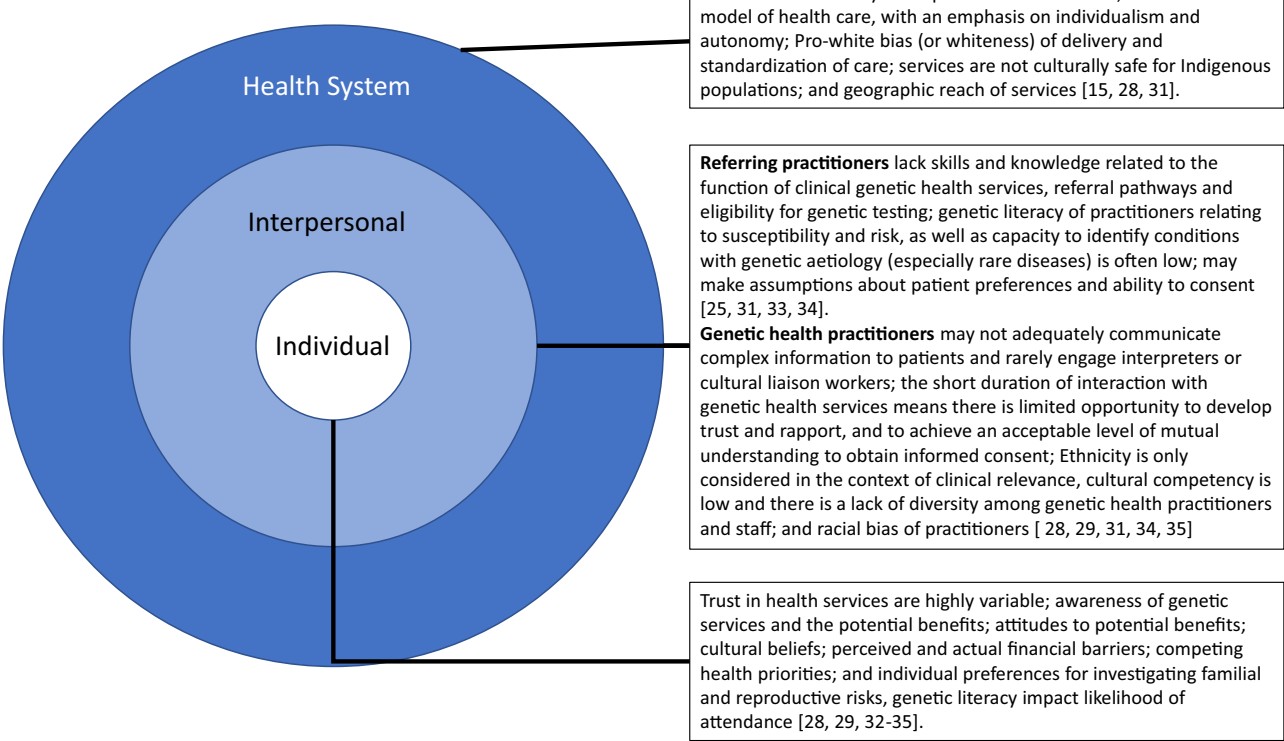

Mainstream health systems promote a Eurocentric, biomedical model of health care, with an emphasis on individualism and autonomy; Pro-white bias (or whiteness) of delivery and standardization of care; services are not culturally safe for Indigenous populations; and geographic reach of services [15, 28, 31].

**Referring practitioners** lack skills and knowledge related to the function of clinical genetic health services, referral pathways and eligibility for genetic testing; genetic literacy of practitioners relating to susceptibility and risk, as well as capacity to identify conditions with genetic aetiology (especially rare diseases) is often low; may make assumptions about patient preferences and ability to consent [25, 31, 33, 34].
**Genetic health practitioners** may not adequately communicate complex information to patients and rarely engage interpreters or cultural liaison workers; the short duration of interaction with genetic health services means there is limited opportunity to develop trust and rapport, and to achieve an acceptable level of mutual understanding to obtain informed consent; Ethnicity is only considered in the context of clinical relevance, cultural competency is low and there is a lack of diversity among genetic health practitioners and staff; and racial bias of practitioners [ 28, 29, 31, 34, 35]

Trust in health services are highly variable; awareness of genetic services and the potential benefits; attitudes to potential benefits; cultural beliefs; perceived and actual financial barriers; competing health priorities; and individual preferences for investigating familial and reproductive risks, genetic literacy impact likelihood of attendance [28, 29, 32-35].

**Fig. 1 | An ecological framework presenting factors described in the literature as associating with lower access to clinical genetic health services for marginalised groups.** Factors identified in the literature as associating with lower access to clinical genetic health services for marginalised groups at the following levels: ☐ Individual ☐ Interpersonal ☐ Health System.

and Torres Strait Islander narratives that are uniquely resonant with concepts of genetics, knowledge transfer and inheritance (e.g. Songlines). Further work has included the partnering with Aboriginal and Torres Strait Islander Controlled Health Organisations that provide physical and telehealth services in remote areas. Recording of Aboriginal and Torres Strait Islander status has also been improved, with all Australian jurisdictions currently ascertaining or moving towards systematic ascertainment of this data in clinical databases. Collectively, these provide a foundation for improved data quality and safety to better address unmet needs, as strengthening reporting will be key to evaluation and improvement of services.

Data presented here reveal marked disparities in access to clinical genetic health services for Aboriginal and/or Torres Strait Islander people in terms of lower rates of appointment scheduling and attendance and should be used to inform interventions to improve access to clinical genetic health services.

## Methods

### Indigenous Peoples participation in research

This research was conducted as part of the Better Indigenous Genomics (BIG) Health Services Study funded by the Lowitja Institute. This was a university led project conducted in partnership with Australian clinical genetic health services. Formal support for this project was provided by Aboriginal Medical Services Alliance Northern Territory, Machado-Joseph Disease Foundation, Bega Garnbirringu Health Service (Kalgoorlie), and the Aboriginal Health Council of Western Australia (via Ethics support). Extensive stakeholder consultation and engagement took place with 14 Aboriginal Health Organisations to identify research study priorities as part of the wider BIG study. The BIG study included a Project Reference Group comprised of government policy makers, academic and clinical experts in genetics and Indigenous health (Australian and international; 11 of 25 members identifying as Aboriginal and Torres Strait Islander people and an

additional 4 members identifying as Indigenous people from other countries). Further, an end-user group of seven Aboriginal women across Australia with lived experience accessing clinical genetic health services and/or working in an Aboriginal organisation informed the research. The authorship included five Aboriginal researchers and clinicians (JL, GG, MJ, YP, GP) who were involved in interpretation of data, including the first author (JL) who designed and conducted the analysis. The research was conducted in accordance with Australian government guidelines for working with Aboriginal people and communities[54].

### Study setting

In Australia, there are eight states and territories, each with its own model of clinical genetic health service provision. We draw on data from three of these: Queensland, Western Australia, and the Northern Territory. Queensland and Western Australia operate a 'hub and spoke' service model, with regular outpatient clinics undertaken in the capital cities of Brisbane and Perth, respectively, and semi-regular outreach clinics held in inner- and outer-regional areas. The Northern Territory operates a fly-in, fly-out service, comprising approximately four blocks of four consulting days per year in the two largest cities, Darwin and Alice Springs. Combined, the three jurisdictions have a resident population of 7.65 million people, roughly a third of the total Australian population. Of these, 5.2% of this population identify as Aboriginal and/or Torres Strait Islander people[55].

### Data sources and linkage

De-identified demographic, clinical and administrative data relating to appointment scheduling and attendance were extracted from the patient database of the state-funded clinical genetic health service in each state: the Northern Territory Genetics Service (Microsoft Excel 2018, for years 2014–2018), Genetic Services of Western Australia (KinTrak, for years 2015–2018), and Genetic Health Queensland

(KinTrak, for years 2015–2017). Patients' Aboriginal status was not recorded in the Genetic Services of Western Australia's internal patient database and was therefore extracted from the overarching hospital service's electronic medical record system (Topaz, King Edward Memorial Hospital, Subiaco) by cross-matching patient hospital record numbers which was recorded in both databases. For population denominators, Census data were retrieved from the Australian Bureau of Statistics (ABS)[55]. Individual consents were not sought as this study was a retrospective analysis of de-identified administrative datasets. Both Clinical Genetic Health Services datasets and Census datasets were available to researchers in de-identified format only. Access and use of these data was in full compliance with local regulatory and legal frameworks governing Clinical Genetic Health Services and census datasets.

### Variable definitions

Data were complete for most variables, except for Aboriginal and/or Torres Strait Islander status and remoteness.

'Aboriginal and/or Torres Strait Islander' denotes people who self-identified (or were identified by a guardian) as Aboriginal and/or Torres Strait Islander either during their clinical genetic health consultation or on intake to the hospital system. A binary indicator of 'Aboriginal and/or Torres Strait Islander person' and 'non- Aboriginal and/or Torres Strait Islander person' was created. We recognise that this dichotomy is a socio-cultural construct that reduces people of over 250 first nations to a single identity. It is a construct that does not represent biological differences, and that both designations represent diverse and genetically heterogenous populations. In Western Australia, 19.7% of patients (2,247 people) did not have Aboriginal and/or Torres Strait Islander status recorded in hospital records and these individuals were excluded from analysis, while this variable was complete for the Northern Territory and Queensland.

Remoteness was coded by matching each patient's residential postcode to the Australian Bureau of Statistics (ABS) recognised Australian Statistical Geography Standard (ASGS) remoteness structure. The ASGS remoteness structure was further coded from five into three categories of – 'Major City'; 'Regional', which included inner and outer regional; and 'Remote', which included remote and very remote ASGS classifications[21].

Age was calculated on the date of a patient's first scheduled appointment from date of birth and stratified into 10-year age groups for analysis. Gender was collected as a binary 'male' or 'female' on basis of self-report (or by guardian).

Appointment location was recorded as in-person at clinic (which included both clinics conducted in major centres and outreach clinics conducted in Aboriginal and/or Torres Strait Islander communities) or as telehealth.

The suspected or known reason for referral to the clinical genetic health service was coded as 'rare disease' or 'cancer'. Prenatal referrals were a subcategory of 'rare disease'.

### Determination of outcomes

Appointment scheduling and appointment attendance were the outcomes of interest. Scheduling of appointment denoted an individual had an appointment scheduled in a clinical genetic health service database. Appointment attendance reflected whether the first scheduled appointment was attended (as opposed to cancelled, re-scheduled, or not attended).

### Statistical analysis

Statistical analysis was conducted using SPSS 27 (SPSS Inc., Chicago, Illinois, USA), with exception of relative risk and rate ratios, which were calculated using Medcalc online (Medcalc software, Ostend, Belgium) and multivariate Poisson regression which was modelled using Stata 16 (StatCorp, College Station, Texas, USA). Microsoft Excel was used to tabulate data including incident appointment scheduling and attendance.

To make comparisons with non-Aboriginal and/or Torres Strait populations, rate ratios were used as a relative measure of disparity and rate difference as an absolute measure. For categorical data, $\chi^2$ tests were used to assess trends in proportions attending appointments attended.

Incidence rates for appointment scheduling were calculated using ABS census data. For population denominators, tables with Aboriginal status, age, gender and state population counts were retrieved from the ABS website[55]. Incidence rates have been presented as annual incidence per 100,000 people as most genetic conditions are individually rare. Rate of attendance was calculated as percent of those with a scheduled appointment who attended.

Multivariate Poisson regression was used to measure disparity in appointment scheduling and attendance. For appointment scheduling, a frequency weight variable was used, and results expressed as an incident rate ratio (IRR). Both models included adjustment for age, gender, and state.

## Data availability

The individual patient level data generated in this study are not able to be shared for privacy and ethical reasons. The raw data from Genetic Health Queensland, Genetic Services of Western Australia, and Northern Territory Genetic Services are available under restricted access due to data privacy laws. These raw data may be accessed through direct application with Genetic Health Queensland, Genetic Services of Western Australia, and Northern Territory Genetic Services. All data provided by clinical genetic health services were pre-processed and provided to researchers with patients names removed. Deidentified data were accessed by JL and PD only.

Data from the Australian Bureau of Statistics is accessible via their website [https://www.abs.gov.au/statistics/people/aboriginal-and-torres-strait-islander-peoples/estimates-aboriginal-and-torres-strait-islander-australians/latest-release#data-download]. The remaining data are available within the Article or from the authors upon request.

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

## Acknowledgements

This paper is dedicated to the memory of Professor Margaret Kelaher. We thank the Lowitja Institute and local Aboriginal Health Organisations for their involvement in extensive stakeholder consultation and engagement activities. We thank Cassie Greer and Rachel Austin for their technical database support. We acknowledge funders including the National Health and Medical Research Council (Australia): 114737 and Lowitja Institute: 1364. J.L. is recipient of a Heart Foundation Australian Indigenous PhD Scholarship—100699. G.G. salary is supported by a NHMRC Investigator Grant (#1176651).

## Author contributions

J.L. designed and performed epidemiological analyses and wrote the manuscript. M.K. co-conceptualised the overall study of which this research forms a part with R.S., as well as input from Y.P. M.K. also led the acquisition of funding and provided statistical advice. P.D. oversaw administration of the project and performed data collection and management. P.D., M.K. and G.B. assisted with drafting the manuscript. The study was facilitated by G.B. in Western Australia, L.T. and C.A.S. in the Northern Territory and A.S.F. in Queensland. R.S., J.Mc., L.M., G.G., E.K., H.D., M.J., Y.P., G.P. and G.B. were involved in design of the study and/or contributed to funding acquisition. All authors reviewed and provided either comments or edits on the manuscript.

## Competing interests

R.S. was the Director of the Northern Territory Genetics Service between 2010–20. CAS commenced as Director of the Northern Territory Genetics Service in 2021. G.B. is a consultant clinical geneticist at Genetics Services of Western Australia and J.Mc. is the Director of Genetic Health Queensland. H.D. was the Director of the Office of Population Health Genomics, Public and Aboriginal Health Division, Government of Western Australia until Nov 2018. The remaining authors declare no competing interests.

## Ethics

This research complies with all relevant ethic regulations. Ethics approval was obtained from the following Human Research Ethics Committees: The University of Melbourne (HREC-1648489.4), Northern Territory Department of Health and Menzies School of Health Research (HREC-2018-3075) and the Central Australian Health Service (HREC-18 3112), The Queensland Department of Health (HREC/18/QTHS/51), the Aboriginal Health Council of Western Australia (HREC-810) and the King Edward Memorial Hospital (RGS0000000513).
