## [Peer Review File · Nature Communications]

Investigating disparity in access to Australian clinical genetic health services for Aboriginal and Torres Strait Islander peopleEditorial note: This manuscript has been previously reviewed at another journal that is not operating a transparent peer review scheme. This document only contains reviewer comments and rebuttal letters for versions considered at Nature Communications.

Peer review comments, first round review –

Reviewer #1 (Remarks to the Author):

For the study now titled “Investigating inequity in access to Australian clinical genetic health services for Aboriginal and Torres Strait Islander people” the authors conclude that “Our analyses provide clear evidence for marked inequity in access for Aboriginal and/or Torres Strait people, both in terms of appointment scheduling and attendance at clinical genetic health services across three jurisdictions.”

In response to the suggestion to provide more clarity of what equity/inequity means, they added that “we can only assume equity is achieved at equal rates of appointment scheduling and recognise that this may underestimate the actual needs of Aboriginal and/or Torres Strait Islander people to access clinical genetic health services.” Further, in response to reviewer 4, it seems, they switched from using the term “equity” to “inequity”, presumably sharing the view of R4 that these terms “are not opposite sides of the same coin – equity comes from philosophy whereas inequity comes more from observational epidemiology.”

As someone working at the intersection of ethics and public health with close familiarity of the respective discourses, I do not share R4’s characterization, and am likewise not convinced that the author’s response solves the issue that they make fairly strong claims about inequitable uptake of services without providing further clarity on

- a) the extent of unmet need (the stronger the unmet need, the more concerning differences in uptake--without such evidence, it is not clear whether we need to be concerned about uptake of generic services)
- b) whether and what kind of policy response is required (equal uptake, higher uptake etc)

The authors note that “this paper uses observational epidemiology”, that 2 qualitative studies with N=62 and N=252 participants “qualify want for services by Aboriginal and Torres Strait Islander populations” even if “we really don’t know the actual need by groups.”

Moreover, in response to provide further clarity on the exact differences in uptake in the 3 areas re scheduled appointments, they now clarify that:

“Aboriginal and/or Torres Strait Islander people comprised 20.4%, 3.3% and 4.2% of all scheduled appointments in Northern Territory, Western Australia, and Queensland respectively, slightly lower than the expected 30.3%, 3.9% and 4.6% we would expect based on population distribution.”

That is we are talking about differences of almost 10%, vs 0.5% and 0.4% (albeit, with, obviously different base %---but the difference is still 3x/5x lower, relative to the almost 50% disparity)

This is clearly an important study showing differences across relevant population groups—but whether these differences constitute inequities, ie, unfair differences, seems far from clear---and the authors made it plain in their responses that they have no plans of further investigating sources of (regional, or other) differences, responding repeatedly that their methods do not permit this.

This being so, the most appropriate way forward would be to completely dial back the claim about inequity and leave out any normative reference points, that is, simply speak about disparities or inequalities, only (in the title and throughout the text). This would retain all of the merits of the study, while not shouldering a burden of proof that is still not met.

Reviewer #3 (Remarks to the Author):

I enjoyed reviewing this paper again having seen it in its previous form for another Journal. I note the reviewer responses to my comments - I was Reviewer 3.

The authors are very honest in terms of the limitations that their study design offer - a cross sectional study limited by the structure and content of pre-existing databases that in many instances offer little granularity in some important data fields. It was gratifying to see that there was substantial consultation and involvement of communities in this project including several ATSI individuals on the authorship team. The authors do refer to this as "governance" but I would tend to differ in this view - governance refers to the ability to exert power and control over a project - where the involvement here seems to me at the level of consultation.

This study offers a baseline of the extent of inequity for provision of genetic services which qualitatively mirrors provision gaps across multiple areas of medicine in Australia. Consequently this paper does not offer an international exemplar of a new direction on how to methodologically explore these issues or offer generalisable insights for other nations. It is undeniably an important start for Australia to improve its performance. The novelty will come with future studies which will involve prospective data aggregation that will reflect the need for information on factors that may point to policy alterations. In this respect this study was an important exploratory exercise which I commend the authors for conducting so well.

Reviewer #4 (Remarks to the Author):

The results are well summarised, clearly identifying the observed findings and giving possible/probable explanations and limitations of the data and analysis.

The research is noteworthy in that it quantifies - for the first time - an analysis of access by an indigenous population (Australian Aboriginal) to genomic health services. Regrettably these results are not novel in the sense that they further quantify the persistent inequities that the Australian health system produces for its indigenous peoples.

The data is generally good; there are weaknesses and reporting biases (which are clearly acknowledged) and publication may well support the improvement of these data into the future.

The cross-sectional data are unable to show causality, yet they can demonstrate the inequity evident and give associations.

One possible shortcoming (and these authors are not alone with this challenge) is that inequities tend to be measured as population comparisons (Aboriginal versus non-Aboriginal patients) and typically different rates of access and poorer outcomes are seen. However, in an area like genomic medicine which is fairly new, novel and not typically universally accessible, it is likely that non-Aboriginal access is also constrained by an (in)adequacy of service provision. This exacerbates access by more vulnerable populations (due to rationing) and also provides an undercount in BOTH the comparison populations. That is, it would be better to judge inequity not only against other populations but against the met/unmet NEEDS that Aboriginal people have for genomic health services. Taken together the inequities are likely to be unquantified underestimates. Examining this limitation would be a valuable addition to the analysis.

Unpacking the mechanisms of inequities is not possible with these data, however the authors show a good command of the extant literature.

[I have reviewed an earlier version and this draft is much improved. The research limitations are managed as well as could be expected].

REVIEWER COMMENTS

Reviewer #1 (Remarks to the Author):

For the study now title "Investigating inequity in access to Australian clinical genetic health services for Aboriginal and Torres Strait Islander people" the authors conclude that "Our analyses provide clear evidence for marked inequity in access for Aboriginal and/or Torres Strait people, both in terms of appointment scheduling and attendance at clinical genetic health services across three jurisdictions."

In response to the suggestion to provide more clarity of what equity/inequity means, they added that "we can only assume equity is achieved at equal rates of appointment scheduling and recognise that this may underestimate the actual needs of Aboriginal and/or Torres Strait Islander people to access clinical genetic health services." Further, in response to reviewer 4, it seems, they switched from using the term "equity" to "inequity", presumably sharing the view of R4 that these terms "are not opposite sides of the same coin – equity comes from philosophy whereas inequity come more from observational epidemiology."

As someone working at the intersection of ethics and public health with close familiarity of the respective discourses, I do not share R4's characterization, and am likewise not convince that the author's response solves the issue that they make fairly strong claims about inequitable uptake of services without providing further clarity on

a) the extent of unmet need (the stronger the unmet need, the more concerning differences in uptake---without such evidence, it is not clear whether we need to be concerned about uptake of generic services)

b) whether and what kind of policy response is required (equal uptake, higher uptake etc)

The authors note that "this paper uses observational epidemiology", that 2 qualitative studies with N=62 and N=252 participants "qualify want for services by Aboriginal and Torres Strait Islander populations" even if "we really don't know the actual need by groups."

In our October 2021 submission we did include further clarification relating to the need for services, but have expanded on this in the latest iteration of the manuscript. We have now referenced national morbidity and population data that demonstrates higher rates of incident cancers, higher fertility rates for Aboriginal and Torres Strait Islander women, and younger population age distribution of the Aboriginal and Torres Strait Islander population. These data suggest that the need for genetic services is likely higher for Aboriginal and Torres Strait Islander people, rather than equal or lower.

"As we consider the need for clinical genetic health services, all available data suggests there is a greater need for services amongst Aboriginal and/or Torres Strait Islander populations. Australian Institute of Health and Welfare data reveals higher rates of incident cancers for Aboriginal and/or Torres Strait Islander populations, suggesting a likely greater need for clinical genetic health services amongst Aboriginal and/or Torres Strait Islander people with cancer and their families²². Similarly, the Australian Bureau of Statistics reports higher fertility rates for Aboriginal and or/Torres Strait Islander women, who, across their lifetime, have an average of 2.32 babies per woman compared to 1.66 babies per non-Aboriginal and or/Torres Strait Islander woman, which also suggests a greater need for clinical genetic health services in the prenatal period²³. A younger age distribution for the Aboriginal and/or Torres Strait Islander population also suggests we should expect a greater frequency of rare diseases amongst Aboriginal populations, as rare diseases are mostly diagnosed early in life²⁴. Further, as we consider that the term 'Aboriginal and/or Torres Strait Islander' is a social construct which includes people of great biological heterogeneity, there is no prima facie reason to expect lower susceptibility to either rare genetic conditions in pregnancy or hereditary cancer for Aboriginal and Torres Strait Islander people. All these factors combined, allude to a greater need for clinical genetic health services rather than equal need for Aboriginal and/or Torres Strait Islander people" (Page 5, Line 21)

This national mortality and population data provides some indication of need while the qualitative studies we referenced provide limited evidence of want. We recognise these qualitative studies are not robust in terms of sample size but these data are all that is available relating to need or want for services. This is an emerging research area.

b) whether and what kind of policy response is required (equal uptake, higher uptake

etc)" We have provided further details (highlighted) regarding a policy response.

“In this paper we have likely underestimated the actual disparity rates Aboriginal and/or Torres Strait Islander people to accessing clinical genetic health services. Our interpretation of data is based on assumed equal need for Clinical Genetic Health Services by both groups. However, based on the higher rates of incident cancer for Aboriginal and Torres Strait Islander people and higher rates of fertility amongst Aboriginal and Torres Strait Islander women, we should have seen incident appointment scheduling rates greater for Aboriginal and/or Torres Strait Islander people rather than equal. Further work is required to quantify with greater precision the actual need and want for clinical genetic health services by Aboriginal and/or Torres Strait Islander people. But it is likely that policy responses to increase access to genetic health services will need to target a higher rate of uptake for Aboriginal and/or Torres Strait Islander populations rather than equal uptake for both populations.” (Page 13 Line 16)

Moreover, in response to provide further clarity on the exact differences in uptake in the 3 areas re scheduled appointments, they now clarify that:

“Aboriginal and/or Torres Strait Islander people comprised 20.4%, 3.3% and 4.2% of all scheduled appointments in Northern Territory, Western Australia, and Queensland respectively, slightly lower than the expected 30.3%, 3.9% and 4.6% we would expect based on population distribution.”

That is we are talking about differences of almost 10%, vs 0.5% and 0.4% (albeit, with, obviously different base %---but the difference is still 3x/5x lower, relative to the almost 50% disparity)

This is clearly an important study showing differences across relevant population groups— but whether these differences constitute inequities, ie, unfair differences, seems far from clear---and the authors made it plain in their responses that they have no plans of further investigating sources of (regional, or other) differences, responding repeatedly that their methods do not permit this.

This being so, the most appropriate way forward would be to completely dial back the claim about inequity and leave out any normative reference points, that is, simply speak about disparities or inequalities, only (in the title and throughout the text). This would retain all of the merits of the study, while not shouldering a burden of proof that is still not met.

We have taken this advice and used the term disparity throughout the paper

Reviewer #3 (Remarks to the Author):

I enjoyed reviewing this paper again having seen it in its previous form for another Journal. I note the reviewer responses to my comments - I was Reviewer 3.

The authors are very honest in terms of the limitations that their study design offer - a cross sectional study limited by the structure and content of pre-existing databases that in many instances offer little granularity in some important data fields. It was gratifying to see that there was substantial consultation and involvement of communities in this project including several ATSI

individuals on the authorship team. The authors do refer to this as "governance" but I would tend to differ in this view - governance refers to the ability to exert power and control over a project - where the involvement here seems to me at the level of consultation.

We have amended "Indigenous governance of research" to "Indigenous peoples participation in research" (page 18 Line 22)

This study offers a baseline of the extent of inequity for provision of genetic services which qualitatively mirrors provision gaps across multiple areas of medicine in Australia. Consequently this paper does not offer an international exemplar of a new direction on how to methodologically explore these issues or offer generalisable insights for other nations. It is undeniably an important start for Australia to improve its performance. The novelty will come with future studies which will involve prospective data aggregation that will reflect the need for information on factors that may point to policy alterations. In this respect this study was an important exploratory exercise which I commend the authors for conducting so well.

Reviewer #4 (Remarks to the Author):

The results are well summarised, clearly identifying the observed findings and giving possible/probable explanations and limitations of the data and analysis.

The research is noteworthy in that it quantifies - for the first time - an analysis of access by an indigenous population (Australian Aboriginal) to genomic health services. Regrettably these results are not novel in the sense that they further quantify the persistent inequities that the Australian health system produces for its indigenous peoples.

The data is generally good; there are weaknesses and reporting biases (which are clearly acknowledged) and publication may well support the improvement of these data into the future.

The cross-sectional data are unable to show causality, yet they can demonstrate the inequity evident and give associations.

One possible shortcoming (and these authors are not alone with this challenge) is that inequities tend to be measured as population comparisons (Aboriginal versus non-Aboriginal patients) and typically different rates of access and poorer outcomes are seen. However, in an area like genomic medicine which is fairly new, novel and not typically universally accessible, it is likely that non-Aboriginal access is also constrained by an (in)adequacy of service provision. This exacerbates access by more vulnerable populations (due to rationing) and also provides an undercount in BOTH the comparison populations. That is, it would be better to judge inequity not only against other populations but against the met/unmet NEEDS that Aboriginal people have for genomic health services. Taken together the inequities are likely to be unquantified underestimates. Examining this limitation would be a valuable addition to the analysis.

We take this comment on board regarding met/unmet needs. However, we are also mindful that ascertaining population need is difficult when it comes to clinical genetic health services. Unlike coronary heart disease where need for angiography in health services is easily identifiable by counting number of people with angina, TIA or stroke etc. It is a bit more complicated for genetic conditions, as genetic conditions are rare and include so many, many different conditions, of varying

health impact. The suggested analysis would not currently be possible without better data that captures need for services amongst Aboriginal and/or Torres Strait Islander people.

Unpacking the mechanisms of inequities is not possible with these data, however the authors show a good command of the extant literature.

[I have reviewed an earlier version and this draft is much improved. The research limitations are managed as well as could be expected].

Peer review comments, second round review –

Reviewer #1 (Remarks to the Author):

it is helpful that the authors were able to identify further data that helps the reader appreciate the the type and magnitude of differences the team is concerned with. The associated normative claims re the is/ought of screening use are now framed far more adequately.

Reviewer #4 (Remarks to the Author):

The authors have gone to very considered lengths to accommodate the various views of the reviewers here, consistent with the available evidence.

I much prefer inequity to disparity because it conveys the idea of unjust inequality or unjust disparity - that is one which should not exist and likely the result of systemic biases in service availability, provision and prioritisation.

However, I see the reviewers are unlikely to agree on this point and the authors are treading a finely balanced path here. Their references to other work in Australia - which does frame differences as inequity - is helpful, ironically a necessary testament to the biases inherent in peer review itself!

One small point, I would have liked the authors to clearly identify which of them is indigenous.